# Impact-driven Exploration with Contrastive Unsupervised Representations

## Abstract

Procedurally-generated sparse reward environments pose significant challenges for many RL algorithms. The recently proposed impact-driven exploration method (RIDE) by Raileanu & Rocktäschel (2020), which rewards actions that lead to large changes (measured by $\ell_2$-distance) in the observation embedding, achieves state-of-the-art performance on such procedurally-generated MiniGrid tasks. Yet, the definition of "impact" in RIDE is not conceptually clear because its learned embedding space is not inherently equipped with any similarity measure, let alone $\ell_2$-distance. We resolve this issue in RIDE via contrastive learning. That is, we train the embedding with respect to cosine similarity, where we define two observations to be similar if the agent can reach one observation from the other within a few steps, and define impact in terms of this similarity measure. Experimental results show that our method performs similarly to RIDE on the MiniGrid benchmarks while learning a conceptually clear embedding space equipped with the cosine similarity measure. Our modification of RIDE also provides a new perspective which connects RIDE and episodic curiosity (Savinov et al., 2019), a different exploration method which rewards the agent for visiting states that are unfamiliar to the agent's episodic memory. By incorporating episodic memory into our method, we outperform RIDE on the MiniGrid benchmarks.

## 1 Introduction

Reinforcement learning (RL) algorithms aim to learn an optimal policy that maximizes expected reward from the environment. The search for better RL algorithms is motivated by the fact that many complex real-world problems can be formulated as RL problems. Yet, environments with sparse rewards, which often occur in the real-world, pose a significant challenge for RL algorithms that rely on random actions for exploration. Sparsity of the reward can make it extremely unlikely for the agent to stumble upon any positive feedback by chance. The agent may spend a long time simply exploring and not receiving a single positive reward.

To overcome this issue of exploration, several previous works have used intrinsic rewards (Schmidhuber, 1991; Oudeyer et al., 2007; 2008; Oudeyer & Kaplan, 2009). Intrinsic rewards, as the name suggests, are reward signals generated by the agent which can make RL algorithms more sample efficient by encouraging exploratory behavior that is more likely to encounter rewards. Previous works have used state novelty in the form of state visitation counts (Strehl & Littman, 2008) for tabular states, pseudo-counts for high-dimensional state spaces (Bellemare et al., 2016; Ostrovski et al., 2017; Martin et al., 2017), prediction error of random networks (Burda et al., 2019b), and curiosity about environment dynamics (Stadie et al., 2015; Pathak et al., 2017) as intrinsic rewards.

Although such advances in exploration methods have enabled RL agents to achieve high rewards in notoriously difficult sparse reward environments such as Montezuma's Revenge and Pitfall (Bellemare et al., 2013), many existing exploration methods use the same environment for training and testing (Bellemare et al., 2016; Pathak et al., 2017; Aytar et al., 2018; Ecoffet et al., 2019). As a result, agents trained in this fashion do not generalize to new environments. Indeed, several recent papers point out that deep RL agents overfit to the environment they were trained on (Rajeswaran et al., 2017; Zhang et al., 2018; Machado et al., 2018), leading to the creation of new benchmarks consisting of procedurally-generated environments (Cobbe et al., 2019; Risi & Togelius, 2020; Küttler et al., 2020).

In practice, agents often have to act in environments that are similar, but different from the environments they were trained on. Hence, it is crucial that the agent learns a policy that generalizes across diverse (but similar) environments. This adds another layer of difficulty, the diversity of environment layout for each episode, to the already challenging sparse reward structure. To tackle this challenge head-on, Raileanu & Rocktäschel (2020) focus on exploration in procedurally-generated environments and propose RIDE, an intrinsic rewarding scheme based on the "impact" of a new observation. Denoting the observation embedding function by $\phi$, RIDE measures the impact of observation $o'$ by computing $\|\phi(o') - \phi(o)\|_2$, where $o$ is the previous observation. Similarly, Savinov et al. (2019) propose episodic curiosity (EC), an intrinsic rewarding scheme which rewards visiting states that are dis-similar to states in the agent's episodic memory.

RIDE uses forward and inverse dynamics prediction (Pathak et al., 2017) to train the observation embedding $\phi$ in a self-supervised manner. Hence, a question that one might ask is:

*What is the $\ell_2$-distance in this embedding space measuring?*

We address this question by modifying the embedding training procedure, thereby changing the definition of impact. That is, we modify RIDE so that impact corresponds to an explicitly trained similarity measure in the embedding space, where we define two observations to be similar if they are reachable from each other within a few steps. The original definition of "impact" in RIDE is not conceptually clear because the learned embedding space is not inherently equipped with a similarity measure, let alone $\ell_2$-distance. It is still possible that RIDE's measure of impact based on $\ell_2$-distance may implicitly correspond to some similarity measure in the embedding space, but we leave this investigation for future work.

Our main contributions are 1) proposing a conceptually clear measure of impact by training observation embeddings explicitly with the cosine similarity objective instead of forward and inverse dynamics prediction, 2) providing a new perspective which connects RIDE and EC, and 3) outperforming RIDE via episodic memory extensions. We use SimCLR (Chen et al., 2020) to train the embedding function and propose a novel intrinsic rewarding scheme, which we name RIDE-SimCLR. As in EC, the positive pairs used in the contrastive learning component of RIDE-SimCLR correspond to pairs of observations which are within $k$-steps of each other (referred to as "$k$-step reachability" in their work).

Following the experimental setup of Raileanu & Rocktäschel (2020), we use MiniGrid (Chevalier-Boisvert et al., 2018) to evaluate our method as it provides a simple, diverse suite of tasks that allows us to focus on the issue of exploration instead of other issues such as visual perception. We focus on the comparison of our approach to RIDE since Raileanu & Rocktäschel (2020) report that RIDE achieves the best performance on all their MiniGrid benchmarks among other exploration methods such as intrinsic curiosity (ICM) by Pathak et al. (2017) and random network distillation (RND) by Burda et al. (2019b). We note that MiniGrid provides a sufficiently challenging suite of tasks for RL agents despite its apparent simplicity, as ICM and RND fail to learn any effective policies for some tasks due to the difficulty posed by procedurally-generated environments. Our experimental results show that RIDE-SimCLR performs similarly to RIDE on these benchmarks with the added benefit of having a conceptually clear similarity measure for the embedding space.

Our qualitative analysis shows interesting differences between RIDE and RIDE-SimCLR. For instance, RIDE highly rewards interactions with controllable objects such as opening a door, which is not the case in RIDE-SimCLR. We also observe that our episodic memory extension improves the quantitative performance of both methods, which demonstrates the benefit of establishing a connection between RIDE and EC. The Never Give Up (NGU) agent by Badia et al. (2020) can be seen as a close relative of our memory extension of RIDE since it uses $\ell_2$ distance in embedding space trained with the same inverse dynamics objective to compute approximate counts of states and aggregates episodic memory to compute a novelty bonus.

Our work is different from EC because we do not explicitly sample negative pairs for training the observation embedding network, and we use cosine similarity, instead of a separately trained neural network, to output similarity scores for pairs of observations. We note that Campero et al. (2020) report state-of-the-art results on even more challenging MiniGrid tasks by training an adversarially motivated teacher network to generate intermediate goals for the agent (AMIGo), but we do not compare against this method since their agent receives full observations of the environment. Both RIDE and RIDE-SimCLR agents only receive partial observations.

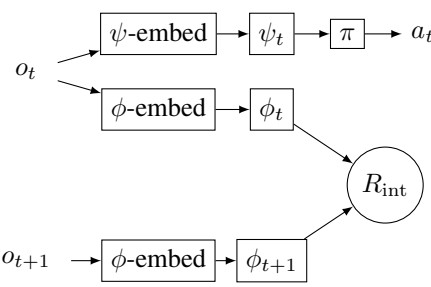
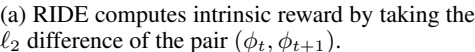

(a) RIDE computes intrinsic reward by taking the $\ell_2$ difference of the pair $(\phi_t, \phi_{t+1})$.

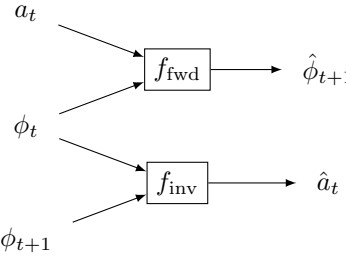

(b) RIDE trains the observation embedding $\phi$ by minimizing forward and inverse dynamics prediction error.

Figure 1: RIDE rewards the agent for taking actions that lead to large impact, which is measured by the change in the observation embedding $\phi$.

## 2 BACKGROUND

We consider the standard episodic RL setting in which an agent interacts with the environment to maximize its expected scalar reward for each episode. The interaction proceeds in discrete time steps and terminates at some fixed time $T$ unless the goal is achieved earlier. At time step $t$, the agent receives observation $o_t$ and samples an action from its policy $\pi(o_t)$. The environment then provides the agent with a scalar reward $r_t$, the next observation $o_{t+1}$, and an end-of-episode indicator. The goal of the agent is to maximize the expected discounted reward $R = \sum_{t=1}^{T} \gamma^t r_t$, where $r_t$ is the (extrinsic) reward given by the environment at time step $t$.

When extrinsic reward is sparse, standard RL algorithms such as PPO (Schulman et al., 2017) or IMPALA (Espeholt et al., 2018) often fail to learn a good policy. To overcome this challenge, previous works have proposed augmenting the reward function by $\hat{r}_t = r_t + w_i r_t^i$, where $r_t^i$ is the intrinsic reward and $w_i$ is a hyperparameter which controls the relative importance of $r_t^i$. The intrinsic reward $r_t^i$ is typically designed to be a dense reward function which pushes the policy towards exploratory behavior more likely to encounter positive extrinsic rewards.

Note that intrinsic rewards can be used with any existing RL algorithms without altering their underlying network architecture or training procedure. It suffices to simply replace the extrinsic reward $r_t$ with the augmented reward $\hat{r}_t$. The observation embedding used to compute intrinsic rewards can be trained either 1) *offline* with respect to a uniformly random policy or 2) *online* with respect to agent's current policy. For a fair comparison with Raileanu & Rocktäschel (2020), we use the embedding trained online for computing intrinsic rewards.

### 2.1 IMPACT-DRIVEN EXPLORATION (RIDE)

RIDE (Raileanu & Rocktäschel, 2020) is an intrinsic reward based on the magnitude of change in the observation embedding produced by the agent's action (See Figure 1). More precisely, RIDE is defined as

$$R_{\text{IDE}} \equiv r_t^i(s_t, a_t) = \frac{\|\phi(o_{t+1}) - \phi(o_t)\|_2}{\sqrt{N_{\text{ep}}(s_{t+1})}} \; , \tag{1}$$

where $\phi$ is the observation embedding function and $N_{\text{ep}}(s)$ is the number of times state $s$ is visited within the current episode. The purpose of this discount by $N_{\text{ep}}(s)$ is to prevent the agent from going back and forth a sequence of states with large $\ell_2$ differences.

The embedding function $\phi(o)$ used in RIDE is parametrized by a neural network and trained by minimizing forward and inverse dynamics prediction error (Pathak et al., 2017). Note that the policy network $\pi$ has its own observation embedding network $\psi$, which is trained separately from $\phi$. The embedding $\phi$ is only used to compute the intrinsic reward and never used for control, and the opposite holds for $\psi$. The purpose of using forward and inverse dynamics models to train the embedding is to store information that is useful for predicting the agent's action or effects actions have on the environment. This leads to learning an action-focused embedding space.

RIDE builds upon the intrinsic curiosity module (ICM) by Pathak et al. (2017), which uses the forward dynamics prediction error as intrinsic reward. The novelty in RIDE is using the $\ell_2$ distance between two *different* observations as a measure of qualitative change between the states. Indeed, visualizations of RIDE by Raileanu & Rocktäschel (2020) show that RIDE assigns higher intrinsic rewards for actions that qualitatively "change the dynamics", such as opening doors or interacting with objects in MiniGrid. However, RIDE introduces conceptual difficulties as the embedding space is not explicitly trained with any similarity measure. That is, the forward and inverse dynamics objective does not explicitly pull together or push apart embeddings of *different* observations. In ICM, the $\ell_2$ distance between $\phi(o_{t+1})$ and the prediction $\hat{\phi}(o_{t+1})$ has a clear interpretation as the forward prediction "error", since the forward dynamics objective explicitly minimizes this $\ell_2$ distance. RIDE, on the other hand, uses the $\ell_2$ distance between different observations $o_t$ and $o_{t+1}$ as the intrinsic reward. Yet, the forward and inverse dynamics prediction objective does not specify which pairs of observation embeddings $(o_i, o_j)$ should be pulled closer together and which should be pushed apart (in $\ell_2$ distance).

This does not mean that RIDE fails to capture qualitative changes in the dynamics. In fact, our visualizations (Figure 7) corroborate the findings of Raileanu & Rocktäschel (2020) by demonstrating that RIDE assigns higher rewards for actions such as opening doors. However, it is difficult to precisely define what having "different dynamics" means, let alone giving a quantitative definition of it. Moreover, the question of *why* the $\ell_2$ distance is larger for pairs of observations corresponding to such actions is not well-understood, and thus, requires further investigation. Without understanding *why*, we cannot guarantee that RIDE will always assign higher rewards to actions we perceive as "significantly changing the dynamics".

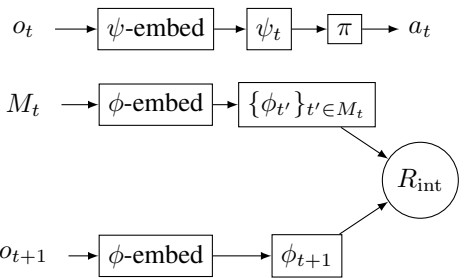 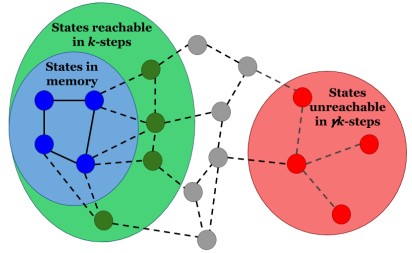

(a) EC computes intrinsic reward by aggregating individual novelty scores in the episodic memory $M_t$.

(b) A state is $k$-step reachable (green) if it can be reached within $k$ steps from states in the current episodic memory (blue).

Figure 2: EC rewards the agent for taking actions that lead to observations different from ones stored in the current episodic memory.

## 2.2 EPISODIC CURIOSITY (EC)

EC (Savinov et al., 2019) is an intrinsic reward based on how "far" the next observation is from observations in the agent's current episodic memory (See Figure 2). More precisely, EC is defined as

$$R_{\mathrm{EC}} \equiv r_t^i(s_t, a_t) = \beta - C(M_t, \phi(o_{t+1})) , \qquad (2)$$

where $\beta \in \mathbb{R}$ is a scalar hyperparameter, $C$ is a learned comparator network, and $M_t$ is the agent's episodic memory at time $t$.

Intuitively, the comparator network $C$ measures how "far" a given observation $o_{t+1}$ is from the agent's current episodic memory $M_t$. The episodic memory $M_t = \{\phi(o_{t'})\}_{t' \leq t}$ is the set of observation embeddings previously encountered in the current episode.[1] The comparator network $C$ predicts whether $o$ and $o'$ are reachable from each other within $k$ steps, i.e., $\hat{y} = C(\phi(o), \phi(o')) \in [0, 1]$ where the true label $y = 1$ if the agent can reach $o'$ from $o$ in $k$ steps and $y = 0$ otherwise. The reachability threshold $k$ is a hyperparameter. The parameters of $C$ and $\phi$ are trained to minimize

---

[1]The definition of episodic memory used in Savinov et al. (2019) is more general. For memory efficiency, they add $\phi(o_{t+1})$ to episodic memory only if it is sufficiently different from the observations already in $M_t$.

CrossEntropy($\hat{y}, y$). The data used in the contrastive training is generated by taking a sequence of the agent's observations $o_1, \ldots, o_N$, and labeling a pair $(o_i, o_j)$ positive if $|i - j| \leq k$ and negative if $|i - j| \geq \gamma k$. Here, $\gamma$ is an additional hyperparameter needed to create a gap between positive and negative examples.

With slight abuse of notation, we write

$$C(M_t, \phi(o_{t+1})) = A(c_1, \ldots, c_{|M_t|}) \in [0, 1] , \tag{3}$$

where $c_i = C(\phi(o_i), \phi(o_{t+1}))$ and $A$ is an aggregation function. The aggregation function pools the individual reachability scores $c_i$ and outputs an aggregate reachability score for $o_{t+1}$ and $M_t$. One simple example of the aggregation function is $A = \max$.

## 3 OUR METHOD

We propose a modification of RIDE to remedy the lack of a conceptually clear similarity measure in the embedding space. Instead of forward and inverse dynamics prediction, we use SimCLR (Chen et al., 2020) to train the observation embeddings with respect to a cosine similarity objective. This has the benefit of equipping the embedding space with a natural similarity measure. We present this direct modification of RIDE in Section 3.1. Our modification opens up a perspective through which we can view RIDE as a special case of EC. From this viewpoint, it is natural to extend RIDE with episodic memory. We present this extension in Section 3.2. A summary of all the methods considered in this work can be found in Table 1.

### 3.1 RIDE WITH CONTRASTIVE UNSUPERVISED REPRESENTATIONS

We propose RIDE-SimCLR, which modifies RIDE by replacing the embedding training involving forward and inverse dynamics models with SimCLR. Denote by $\phi$ the embedding network learned via SimCLR. We define RIDE-SimCLR as

$$R_{\text{SimCLR}} \equiv r_t^i(s_t, a_t) = \frac{1 - \cos(\phi(o_t), \phi(o_{t+1}))}{2\sqrt{N_{\text{ep}}(s_{t+1})}} \in [0, 1] . \tag{4}$$

SimCLR trains representations by maximizing agreement between different perturbations of the same sample (referred to as the "anchor" sample). Given a sequence of consecutive agent observations $o_1, \ldots, o_N$, we perturb each state in two different ways by taking random number of steps sampled from $\{1, \ldots, k\}$ into the future and past (See Figure 3). This gives us $2N$ samples $o'_1, \ldots, o'_{2N}$ among which there are $N$ positive pairs. Note that reachability is dependent on the current behavior policy $\pi$. Unlike EC, however, SimCLR does not explicitly sample negative pairs. Instead, the $2(N - 1)$ remaining samples are treated as negative examples. The loss function, referred to as *NT-Xent* (normalized temperature-scaled cross entropy) in Chen et al. (2020), for a positive pair $(o'_i, o'_j)$ is defined as

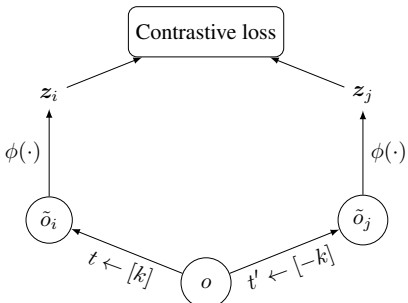

Figure 3: Two time steps $t$ and $t'$ are independently sampled from the set $\{0, \ldots, k\}$, and the input observation is perturbed temporally to generate two correlated views. The embedding network $\phi(\cdot)$ is trained to maximize agreement.

$$\ell(i, j) = -\log \frac{\exp(\cos(\phi(o'_i), \phi(o'_j))/\tau)}{\sum_{m \neq i} \exp(\cos(\phi(o'_i), \phi(o'_m))/\tau)} ,$$

where $\tau$ is the hyperparameter temperature. The total contrastive loss is

$$L = \frac{1}{2N} \sum_{m=1}^{N} \ell(2m - 1, 2m) + \ell(2m, 2m - 1) .$$

We minimize $L$ with respect to the parameters of $\phi$ online. Because $\phi$ is explicitly trained to maximize the cosine similarity between positive pairs, cosine similarity becomes a natural similarity

measure for the learned embedding space. We set the temperature scale $\tau = 0.1$, the reachability parameter $k = 2$, and batch size $N = L/(4k)$ where $L$ is the unroll length of the policy gradient algorithm used to train the policy $\pi$.

## 3.2 EPISODIC MEMORY EXTENSIONS

RIDE-SimCLR can be seen as a special case of EC by setting $C(o_i, o_j) = (1 - \cos(\phi(o_i), \phi(o_j)))/2$, $A(c_1, \ldots, c_t) = c_t/\sqrt{N_{ep}(s_{t+1})}$ in Eq. (3).[2] Note that $N_{ep}(s_{t+1}) = |\{i \in [t] \mid c_i = 0\}|$. This perspective allows us to consider variations of RIDE and RIDE-SimCLR that use different aggregation methods on the episodic memory. Hence, we propose RIDE-MinAgg and EC-SimCLR which aggregate values from the episodic memory with $A = \min$ instead of the discount by visitation count. RIDE-MinAgg is RIDE with $A(c_1, \ldots, c_t) = \min_{i \in [t]} c_i$, and $\phi$ trained with forward and inverse dynamics prediction, and EC-SimCLR is RIDE-SimCLR with $A(c_1, \ldots, c_t) = \min_{i \in [t]} c_i$.

| Method | $\phi$ **trained via** | **Aggregator** $A(c_1, \ldots, c_t)$ |
|---|---|---|
| RIDE | Forward/inverse dynamics | $c_t/\sqrt{N_{ep}(s_{t+1})}$ |
| RIDE-MinAgg | Forward/inverse dynamics | $\min_{i \in [t]} c_i$ |
| RIDE-SimCLR | SimCLR | $c_t/\sqrt{N_{ep}(s_{t+1})}$ |
| EC-SimCLR | SimCLR | $\min_{i \in [t]} c_i$ |

Table 1: Summary of intrinsic rewarding schemes considered in this work.

## 4 EXPERIMENTS

We evaluate our methods on procedurally-generated environments from MiniGrid. We compare our methods against RIDE, which achieves previous state-of-the-art performance on the MiniGrid benchmarks. We report the median and max/min of the average episode return across 3 random seeds. The average episode return is computed as a rolling mean of the previous 100 episodes. We also evaluate RIDE and RIDE-SimCLR on the first level of Mario (Kauten, 2018) using only intrinsic rewards.

### 4.1 ENVIRONMENT

In all our MiniGrid environments, the dynamics is deterministic and agent observation is partial. The agent's view (the highlighted part in Figure 4) is limited to a $7 \times 7$ square centered at the current location, and the agent cannot see through walls or closed doors. There are seven actions the agent can choose from: turn left or right, move forward, pick up or drop an object, toggle, and done.

We evaluate our methods on the following 5 benchmarks. Key-CorridorS3R3, KeyCorridorS4R3, KeyCorridorS5R3, MultiRoomN7S8, MultiRoomN12S10, and MultiRoomN7S4-NoisyTV. The NoisyTV variant implements a "noisy TV" in the MultiRoom environment with a ball that changes color whenever the special "switch channel" action is executed by the agent. Noisy TVs are known to cause problems for curiosity-based exploration methods by turning the agent into a "couch potato" (Burda et al., 2019a). The purpose of this task is to demonstrate that our methods do not suffer from this issue. Further details on the MiniGrid tasks can be found in Appendix B.

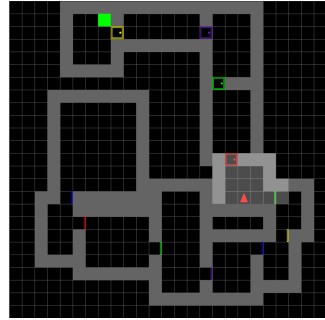

Figure 4: MultiRoomN12S10. The agent (red) has to reach the goal (green) by passing through multiple rooms.

---

[2]Here, we are considering a slightly more general formulation of EC where the comparator network $C$ "absorbs" the constant hyperparameter $\beta$ and the negative sign in Eq. (2). In this case, the value output by $C$ is proportional to dis-similarity rather than similarity.

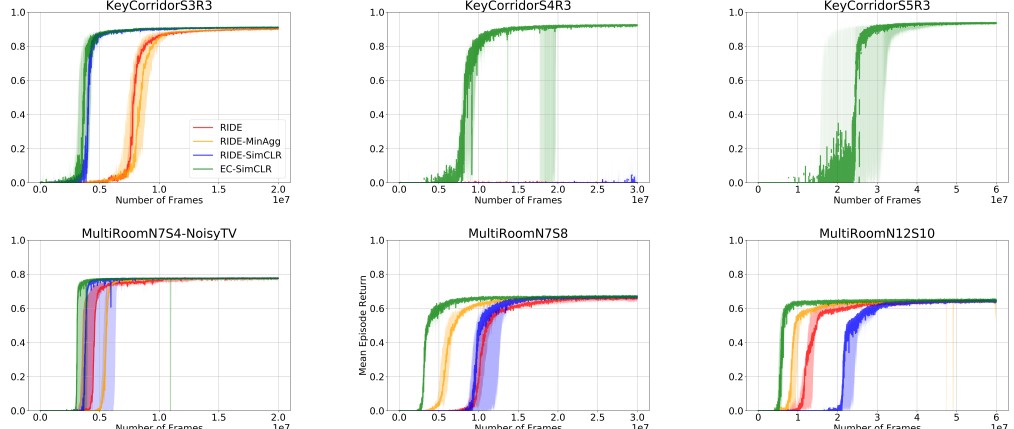

Figure 5: Performance of exploration methods on diverse MiniGrid tasks. Note that EC-SimCLR, the episodic memory extension of RIDE-SimCLR, performs the best on all tasks.

## 4.2 EVALUATION

We evaluate the following four methods: RIDE, RIDE-SimCLR, RIDE-MinAgg, and EC-SimCLR. We use torchbeast (Küttler et al., 2019), an open-source implementation of IMPALA (Espeholt et al., 2018), as the base RL algorithm common to all the methods we evaluate and RMSProp (Tieleman & Hinton, 2012) as the optimizer. All methods we evaluate use the same policy and value network architecture. Their only difference is the intrinsic reward. More details about our network architecture and hyperparameter settings can be found in Appendix A and C.

## 5 RESULTS

### 5.1 QUANTITATIVE RESULTS

Figure 5 shows our results on the MiniGrid benchmarks. RIDE-SimCLR performs similarly to RIDE on all the tasks except MultiRoomN12S10, in which it requires more frames to learn the optimal policy. We note that other exploration methods such as ICM and RND fail to learn any useful policies on this task (Raileanu & Rocktäschel, 2020). Hence, RIDE-SimCLR retains the quantitative performance of RIDE while having a conceptually clear measure for "impact". Moreover, RIDE-MinAgg and EC-SimCLR are more sample-efficient than RIDE on more challenging tasks such as MultiRoomN7S8 and MultiRoomN12S10. In fact, EC-SimCLR is the only method that learns a good policy for KeyCorridorS4R3 and KeyCorridorS5R3. Note, however, that we only run EC-SimCLR on KeyCorridorS5R3 since all other methods have failed in the easier KeyCorridorS4R3 environment and Campero et al. (2020) report that RIDE fails to learn any useful policy on KeyCorridorS4R3 and KeyCorridorS5R3. This demonstrates the benefit of establishing a connection between RIDE and EC. Viewing RIDE through the lens of episodic memory leads to variants of the intrinsic reward that are more sample-efficient.

### 5.2 LEARNED EMBEDDING SPACE

We compare the observation embeddings learned using RIDE-SimCLR and RIDE. Denote by $d(o_i, o_j)$ the dis-similarity measure used in each method. $d$ corresponds to the $\ell_2$ distance between $\phi(o_i)$ and $\phi(o_j)$ in RIDE, and to the cosine dis-similarity $1 - \cos(\phi(o_i), \phi(o_j))$ in RIDE-SimCLR. We analyze how predictive $d(o_i, o_j)$ is for the temporal distance between these two observations. To this end, we generate a balanced labelled dataset consisting of close observation pairs ($\leq k$ steps) and far pairs ($> \gamma k$ steps) by running the trained RIDE agent policy for 100 episodes. The parameters were set to $k = 2$ and $\gamma = 5$. More details on the dataset can be found in Appendix D.

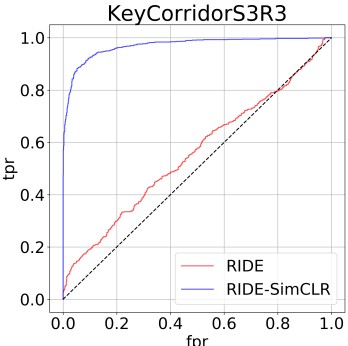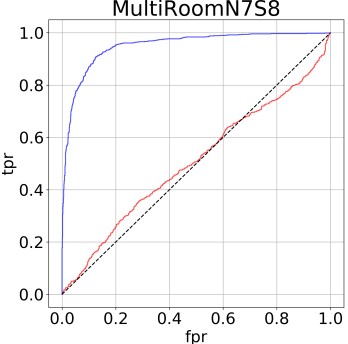

Figure 6: The ROC curves show that the cosine similarity in RIDE-SimCLR is predictive of temporal distance between states, as expected. Unsurprisingly, the $\ell_2$ distance in RIDE is not.

Figure 6 shows the ROC curve of RIDE-SimCLR and RIDE. From the figure, we can see that the cosine similarity measure in RIDE-SimCLR is predictive of the temporal distance between observations. On the other hand, the $\ell_2$ distance between representations from RIDE is not aligned with the temporal distance. This is not surprising since the embedding space in RIDE-SimCLR is explicitly trained to respect temporal distance between observations and RIDE is not. However, this shows that the structure of learned embeddings in RIDE-SimCLR is conceptually clear: temporally close observation are closer in cosine distance. Furthermore, it demonstrates that the intrinsic rewarding schemes of RIDE and RIDE-SimCLR, despite its apparent similarity, are qualitatively different.

|  | Open Door | | Turn Left / Right | | Move Forward | |
|---|---|---|---|---|---|---|
| Model | Mean | Std | Mean | Std | Mean | Std |
| RIDE | 0.0163 | 0.0025 | 0.0064 | 0.0039 | 0.0047 | 0.0022 |
| RIDE-SimCLR | 0.0721 | 0.0107 | 0.0883 | 0.0158 | 0.0545 | 0.0174 |

Table 2: Mean intrinsic reward per action over 100 episodes of MultiRoomN7S8.

### 5.3 INTRINSIC REWARD VISUALIZATION

To understand the qualitative difference between RIDE-SimCLR and RIDE, we analyze which actions are encouraged by each method. We take RIDE-SimCLR and RIDE agents trained on procedurally-generated MultiRoomN7S8 environments and roll-out the learned policies on a fixed environment. Table 2 shows the average intrinsic reward received by each action and Figure 7 shows a heatmap of how the actions were rewarded in the agents' trajectories. We also plot how the average intrinsic rewards change during training in Figure 10 of Appendix G. In Figure 11 of Appendix H, we provide a heatmap that shows how actions were rewarded after RIDE was trained on KeyCorridorS4R3 for 30M frames. Note that RIDE *failed* to learn a good policy for this hard environment. Hence, Figure 11 visualizes a failure mode of RIDE's embedding training.

The results in Table 2 and Figure 7 demonstrate qualitative differences between RIDE-SimCLR and RIDE. As observed by Raileanu & Rocktäschel (2020), RIDE gives higher intrinsic reward for interaction with objects such as opening doors, whereas RIDE-SimCLR gives higher rewards for turning left or right. An intuitive explanation for RIDE is that actions such as "open door" significantly change the dynamics, which in turn leads to a substantial change in RIDE's action-focused embedding. On the other hand, RIDE-SimCLR rewards agents for moving away from where it has been, so actions that move the agent into a new room (which substantially changes the ego-centric partial view of the agent), are given higher rewards. Further investigations into why RIDE and RIDE-SimCLR assign high rewards to these actions is an interesting direction left for future work.

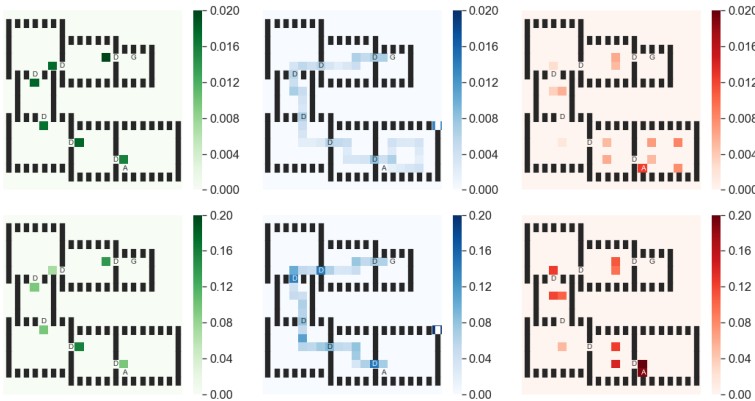

Figure 7: Intrinsic reward heatmaps for opening doors (green), moving forward (blue), or turning left/right (red) on a random instance of MultiRoomN7S8. The top row corresponds to RIDE and the bottom corresponds to RIDE-SimCLR. $A$ is the agent starting position, $G$ is the goal position and $D$ are doors.

### 5.4 EXPLORATION WITH NO EXTRINSIC REWARD

We analyze the exploration behavior of the RIDE-SimCLR agent in the absence of extrinsic reward. We train a RIDE-SimCLR agent on procedurally-generated MultiRoomN10S6 environments for 50M frames with only intrinsic reward as the signal. The agent is allowed to take 200 steps in every episode. We observe that even without any extrinsic reward, the RIDE-SimCLR agent learns a policy which explores all the rooms in the map, similar to the RIDE agent. Other agents trained with intrinsic rewards such as Count, ICM and RND are known to fall short of reaching the final room within the given amount of steps (Raileanu & Rocktäschel, 2020). The state visitation heatmap on a random instance of MultiRoomN10S6 can be found in Appendix E.

Moreover, we compare RIDE-SimCLR to RIDE on the first level of Mario without extrinsic reward to determine its relative performance on environments different from MiniGrid. The results can be found in Appendix F. Our results show that RIDE-SimCLR matches the performance of RIDE on Mario. Note, however, that this singleton environment is not very challenging since even vanilla IMPALA is able to learn similarly good policies without any intrinsic rewards, although IMPALA does use extrinsic rewards (Raileanu & Rocktäschel, 2020).

## 6 CONCLUSION AND FUTURE WORK

We identify a conceptual issue in RIDE and remedy it by learning an observation embedding space naturally equipped with the cosine similarity measure. By training embeddings with SimCLR, we retain the strong performance of RIDE on procedurally-generated MiniGrid benchmarks while getting a conceptually clear similarity measure for the embedding space. Moreover, we make a connection between RIDE and EC. As a result, we outperform both RIDE and RIDE-SimCLR by changing the episodic memory aggregation function, which demonstrates the benefit of this novel perspective.

Despite the apparent similarity between RIDE and RIDE-SimCLR, our analysis shows that these methods are qualitatively different. The $\ell_2$ distance in RIDE, perhaps unsurprisingly, is not predictive of temporal distance between observations, unlike the cosine similarity in RIDE-SimCLR. In addition, actions that are encouraged by each intrinsic rewarding scheme is different. It is possible that $\ell_2$ distance in the embedding space learned with forward and inverse dynamics prediction corresponds to *some* notion of similarity. An interesting future work would be to theoretically and empirically investigate what the $\ell_2$ distance captures in this embedding space. For instance, what does a large $\ell_2$ distance in this space correspond to? Other interesting directions include making intrinsic reward computation from episodic memory time and space efficient using techniques such as $k$-nearest neighbors, as was done in Badia et al. (2020), and proposing variants of EC by training embedding networks with an objective that *implicitly* trains a known similarity measure.

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

## A  NETWORK ARCHITECTURE

All models have the same architecture for the policy and value network. The policy and value network share the same observation embedding network $\psi$. The embedding network consists of 3 convolutional layers with 3x3 kernels, stride of 2, padding of 1, 32 channels, and uses ELU non-linearity (Clevert et al., 2016). The output of $\psi$ is then fed into an LSTM cell (Hochreiter & Schmidhuber, 1997) with 256 units. Two separate fully-connected layers are used on top of the LSTM output to predict the value and action, respectively.

The other embedding network $\phi$ uses the same architecture as $\psi$. Note that $\phi$ is only used to compute the intrinsic reward and its output is never given as an input to the policy network. The opposite is true for $\psi$.

## B  MINIGRID

MiniGrid is a simple and lightweight gridworld gym environment. For all the tasks considered in this work, the dynamics is deterministic and agent observation is partial. The agent's view is limited to a 7x7 square centered at its current location, and the agent cannot see through walls or closed doors. An observation is given as a 7x7x3 tensor. Note that these values are not pixels. It is a partially observable view of the environment using a compact encoding, with 3 input values per visible grid cell. There are seven actions the agent can choose from: turn left or right, move forward, pick up or drop an object, toggle, and done. The agent can pick up and carry exactly one object (e.g., ball or key). To open a locked door, the agent has to be carrying a key matching the door's color. The extrinsic reward given by the environment when goal is reached is $r_t = 1 - 0.9 \cdot (t/t_{\max})$, where the maximum episode length $t_{\max}$ depends on the task.

The following are descriptions of MiniGrid tasks we use for experiments.

- *KeyCorridorSXRY*: the agent has to pick up an object behind a locked door. The key is hidden in another room and the agent has to explore the map to find it. The maximum episode length is $t_{\max} = 30 \cdot X^2$.
- *MultiRoomNXSY*: the agent must navigate through a procedurally-generated map consisting of $X$ many rooms, each of size at most $Y$, to reach the goal. The maximum episode length is $t_{\max} = 20 \cdot X$.
- *MultiRoomNXSY-NoisyTV*: the agent must navigate through a procedurally-generated map consisting of $X$ many rooms, each of size at most $Y$, and a "noisy TV" to reach the goal. The "noisy TV" is implemented with a ball that changes color whenever the special "switch channel" action is executed by the agent. The maximum episode length is $t_{\max} = 20 \cdot X$.

## C  HYPERPARAMETERS

The hyperparameters in Table 3 are common to all the experiments with one exception. For RIDE-SimCLR on MultiRoomN12S10, we used learning rate 0.00005 because the network did not learn any useful policies with learning rate 0.0001. We decay the learning rate linearly so that it becomes 0 after the final epoch.

**RIDE/RIDE-MinAgg**  We used the same set of hyperparameters for RIDE and RIDE-MinAgg. As was done in Raileanu & Rocktäschel (2020), we used an intrinsic reward coefficient 0.1 and entropy coefficient 0.0005 for MultiRoomN7S4-NoisyTV and KeyCorridorS3R3. We used an intrinsic reward coefficient 0.5 and entropy coefficient 0.001 for MultiRoomN7S8 and MultiRoomN12S10. For all tasks, we used a batch size of 32.

| Parameter | Value |
|---|---|
| Learning Rate | 0.0001 |
| Unroll Length | 100 |
| Discount | 0.99 |
| RMSProp Momentum | 0.0 |
| RMSProp $\epsilon$ | 0.00001 |
| Clip Gradient Norm $\epsilon$ | 40.0 |

Table 3: Common hyperparameters

**RIDE-SimCLR/EC-SimCLR** For both methods, we ran a grid search over intrinsic reward coefficient $\in \{0.5, 0.1, 0.05, 0.025, 0.01\}$ and entropy coefficient $\in \{0.001, 0.0005, 0.0001\}$ for all tasks. We used a batch size of 8 for both methods on all tasks.

For RIDE-SimCLR, we used intrinsic reward coefficient 0.01 and entropy coefficient 0.0005 for MultiRoomN7S4-NoisyTV and KeyCorridorS3R3, intrinsic reward coefficient 0.05 and entropy coefficient 0.0005 for MultiRoomN7S8, and intrinsic reward coefficient 0.01 and entropy coefficient 0.0001 for MultiRoomN12S10. For KeyCorridorS4R3, we used intrinsic reward coefficient 0.025 and entropy coefficient 0.0005.

For EC-SimCLR, we used intrinsic reward coefficient 0.01 for MultiRoomN7S4-NoisyTV and KeyCorridorS3R3, 0.05 for MultiRoomN7S8, 0.025 for MultiRoomN12S10, KeyCorridorS4R3, and KeyCorridorS5R3. We used entropy coefficient 0.0005 for all tasks.

## D   ROC Curve Data Generation

We first construct a balanced dataset of size 2000 by sampling 20 pairs per policy roll-out for a total of 100 roll-outs. where each roll-out is performed on a random instance of MultiRoomN7S8. We set hyperparameters $k = 2$ and $\gamma = 5$, and use a policy trained with RIDE on MultiRoomN7S8. We repeat the following procedure 10 times for each roll-out.

1. Given a roll-out $\tau = \{o_1, \ldots, o_T\}$, randomly sample an anchor observation $o_t$.
2. Generate a positive pair $(o_t, o_p)$ by randomly sampling $o_p$ where $p \in [t - k, t + k]$.
3. Generate a negative pair $(o_t, o_n)$ by randomly sampling $o_n$ where $n \notin [t - \gamma k, t + \gamma k]$.
4. Compute data points $x_p = d(o_t, o_p)$ and $x_n = d(o_t, o_n)$. Assign label 0 to $x_p$ and 1 to $x_n$.

We note that this positive/negative pair generation procedure was used by Savinov et al. (2019) to train their reachability network in EC. The parameter $\gamma$ in Step 3 can be thought of a gap which separates the positive samples and negative samples.

## E   No Extrinsic Reward Heatmap

RIDE-SimCLR is able to efficiently explore the state space without any extrinsic reward. Note the stark contrast with purely random exploration, which fails to even leave the first room within the given amount of time steps.

## F   No Extrinsic Reward on Mario

We also compare RIDE-SimCLR to RIDE on the first level of Mario without extrinsic reward to see if RIDE-SimCLR can match RIDE's performance on environments different from MiniGrid. As we can observe from Figure 9, RIDE-SimCLR matches the performance of RIDE on Mario. Note, however, that this singleton environment is not very challenging since even vanilla IMPALA is able to learn similarly good policies without any intrinsic rewards, although it does use extrinsic rewards (Raileanu & Rocktäschel, 2020).

**Random**                                              **RIDE-SimCLR**

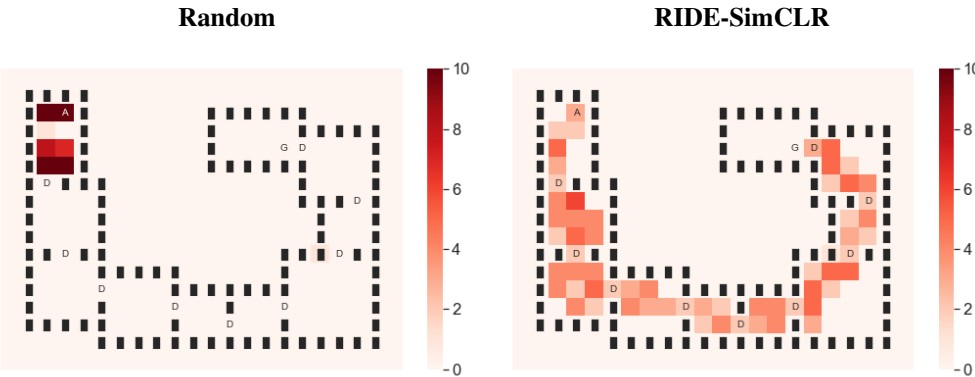

Figure 8: Heatmap of state visitation count on the MultiRoomN10S6 task with a random policy and policy trained with RIDE-SimCLR in the absence of any extrinsic reward.

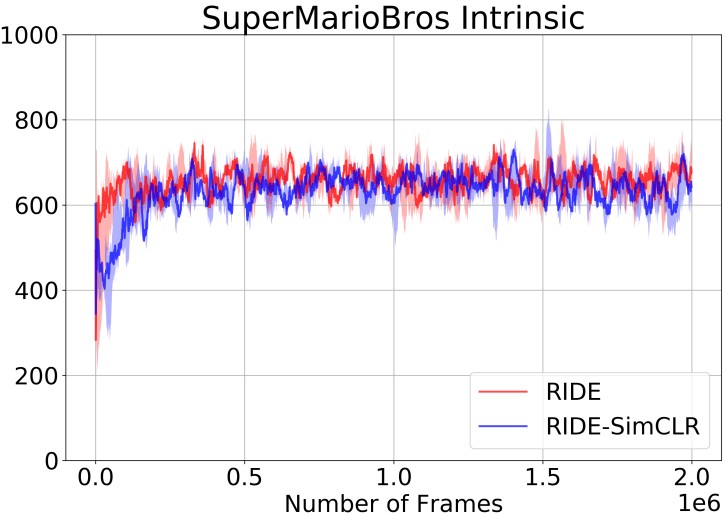

Figure 9: Performance on level 1 of Mario with intrinsic reward only.

The common hyperparameters we used for Mario are the same as ones reported in Table 3, except for the unroll length which we set to 20. For both RIDE and RIDE-SimCLR, we used intrinsic reward coefficient of 0.05 and entropy coefficient of 0.0001.

## G MEAN INTRINSIC REWARD DURING TRAINING

Figure 10 shows the changes in mean intrinsic rewards during training for MultiRoomN7S8, Multi-RoomN12S10, KeyCorridorS3R3, KeyCorridorS4R3.

## H VISUALIZATION OF RIDE ON KEYCORRIDORS4R3

We visualize the intrinsic reward per action of RIDE for KeyCorridorS4R3 in Figure 11. Note that RIDE *failed* to learn a useful policy for this environment (See Figure 5). The purpose of Figure 11 is to visualize the embeddings learned via forward and inverse dynamics when the policy is far from optimal. We can see that the "open door" actions are rewarded less compared to Figure 7 (when RIDE learned a good policy for the environment), and unnecessary turning actions are highly rewarded.

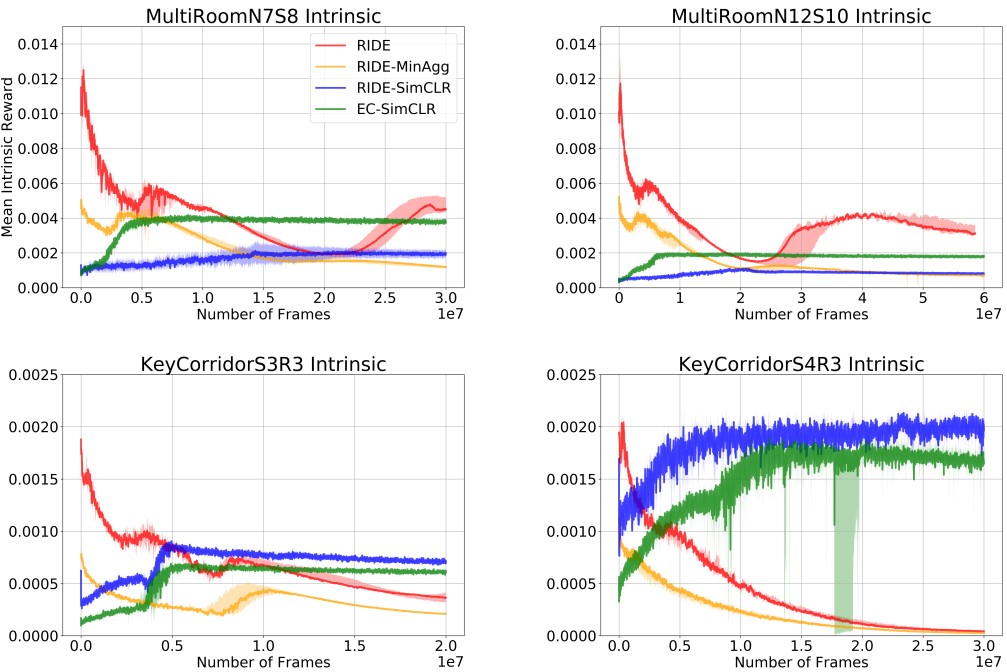

Figure 10: Intrinsic reward during training on diverse MiniGrid tasks.

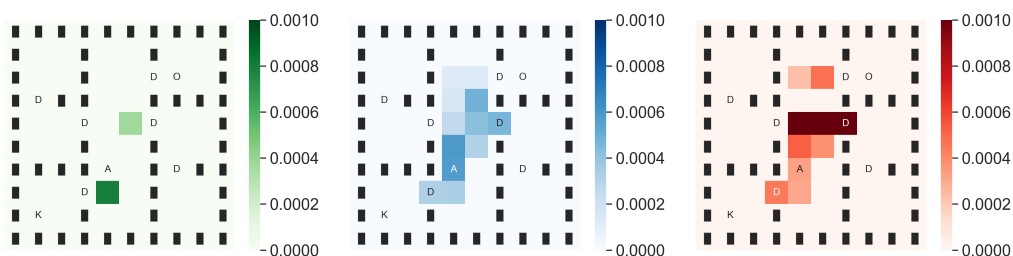

Figure 11: RIDE intrinsic reward heatmaps for opening doors (green), moving forward (blue), or turning left/right (red) on a random instance of KeyCorridorS4R3. $A$ is the agent starting position, $G$ is the goal position and $D$ are doors.

