# OpenReview forum: "Impact-driven Exploration with Contrastive Unsupervised Representations"
_ICLR.cc/2021/Conference — Reject_

### Official Review · AnonReviewer4 · 2020-10-26
**Nice method, with improved exposition after rebuttal**

**Rating:** 7
**Confidence:** 3

**Review:**

## Summary

The work provided a nice new method with some performance gains by combining several existing techniques. The presentation was clear and organized, with the new method getting both better performance and some improvements in interpretability. It provides a variety of visual analyses that are typical of this area of research and present the contrasts between this work and prior efforts.

However, I was unimpressed with the characterization of the RIDE method upon which this method was based.  The paper states:
"The novelty in RIDE is using the `2 distance between two different observations as a
measure of qualitative change between the states. However, this introduces conceptual difficulties
as the embedding space is not explicitly trained with any similarity measure. This is in contrast to
ICM, which trains the state representation to minimize the forward prediction error. Because of this
explicit training objective, there is no conceptual difficulty in viewing the intrinsic reward in ICM
as a measure of its dynamics model uncertainty."

RIDE also trains using prediction errors based on 1) prediction the next state's embedding and 2) predicting the action taken using the two states' embeddings. This encourages the embedding space to store information which can predict actions as well as the effects actions have on the environment. While I admit there is still some uncertainty in this interpretation, the authors of this work do not comment on this explanation at all which should be critical when replacing the mechanism for training these embeddings with an alternative which contains different information. This is quite a big change in the algorithm, as the focus on the impact of actions for the embedding space appears to be the inspiration for calling RIDE "Impact-Driven Exploration".

Given the replacement of action-focused embeddings with distance-focused ones (according to a policy), the author's algorithm might more accurately be called "Depth-Driven" rather than impact-driven.

Before publication, I believe the authors should reduce the emphasis a bit on not being able to explain RIDE's l2 distance (e.g. "This is in contrast to RIDE, in which the l`2 distance is not known to correspond to any similarity measure.") and should recognize the explanation that RIDE gives for its embedding space more clearly so as to be able to more effectively explain the difference (advantage! in these tasks) in their own approach.

## Quality
The paper is presented according to a high standard of quality. They document and explain their methods clearly, including hyperparameters.

## Clarity
The paper presented its results clearly, using the standard visuals for the field and with clearly written and easily readable text. Thank you!

## Originality
The paper provides a novel recombination of methods that others have developed in a non-trivial way with original analysis into the distinctions between these methods and the effects of their combination.

## Significance
The new method shows some improvements over existing methods in terms of sample efficiency on a number of tasks.
The new method has some advantages in interpretability that were explored, though the practical value of this interpretability could be explored a bit further than just sample-efficiency. Are there any kinds of tasks your method might solve that would not be possible with the existing method? For example, you could imagine that a task with extremely long hallways would receive no intrinsic reward in RIDE but your method would have no trouble finding the motivation to keep running down the hallway. In contrast, are there any tasks RIDE would do better at than your approach, or does the "distance"-focused metric work in a strictly superior way? I would have been more impressed by the paper if additional tasks such as this would have been included.

I think a bit less emphasis could be given to the description of RIDE vs RIDE-SimCLR's similarity to temporal distance between states. RIDE is clearly not designed for that, RIDE-SimCLR is, so while it is nice to prove that RIDE-SimCLR does have this property it could be made a tiny bit more brief.

The qualitative comparison with RIDE in terms of the heatmap especially could have been developed a bit more. The authors only say "Why RIDE and RIDE-SimCLR assign high rewards to these actions is an interesting question left
for future work". For RIDE, the reward assignment feels fairly clear: taking the "open door" action creates a substantial change in RIDE's action-focused embedding space, so this is well-rewarded by RIDE. It would be nice for the author's to suggest a contrast with RIDE-SimCLR: it seems to me that it naturally does not provide a lot of intrinsic reward for actions but rather through movement away from where it has been, so it is natural that there is more reward for moving into new rooms rather than specifically on opening the door. RIDE-SimCLR also seems to avoid wandering the first room, which may be because RIDE is looking for objects it could interact with but RIDE-SimCLR just wants to get away from its current position as fast as possible.

Furthermore, I would have appreciated seeing a bit more analysis on running this algorithm without any external reward. This was mentioned in the appendix briefly but the current method was only compared with a random policy and not with RIDE.

---

> ### Author Response · Authors · 2020-11-20
> **Author Response (part 1)**
>
> We thank the reviewer for compliments on our presentation and contributions, and suggesting several ways to improve our exposition.
>
> **"Unimpressed with the characterization of the RIDE ... should recognize the explanation that RIDE gives for its embedding space more clearly"**
>
> We are especially grateful for this comment. We agree that our previous exposition did not sufficiently cover the key insight behind RIDE, so we updated our submission to reflect the reviewer's comments (Section 2.1). In particular, we added the sentences, "The purpose of using forward and inverse dynamics models to train the embedding is to store information that is useful for predicting the agent's action or effects actions have on the environment. This leads to learning an action-focused embedding space."
>
> We also updated our exposition in Section 2.1 to add clarity to our central argument. In particular, we point out the forward and inverse dynamics training objective does not explicitly pull together or push apart embeddings of *different* observation pairs ($o_i, o_j$). To elaborate, in [ICM; 1], the $\ell_2$ distance between $\phi(o_{t+1})$ and the prediction $\hat{\phi}(o_{t+1})$ has a clear interpretation as the forward prediction "error", since the forward dynamics objective explicitly minimizes this $\ell_2$ distance. RIDE, on the other hand, uses the $\ell_2$ distance between *different* observations $o_t$ and $o_{t+1}$ as the intrinsic reward. Yet, the forward and inverse dynamics objective does not specify which pairs of observation embeddings ($o_i$,$o_j$) should be pulled closer together and which should be pushed apart (in $\ell_2$ distance).
>
> However, this does not mean that RIDE fails to capture qualitative change in the dynamics. We expand on this part when addressing the reviewer's next comment, "practical value of (RIDE-SimCLR's) interpretability".
>
> **"Practical value of this interpretability"**
>
> We would like to argue that interpretability itself has practical value as it guarantees that we will get the desired latent structure as long as we train with SimCLR. Since we explicitly specify which observation pairs should be pulled closer together and which pairs should be pushed apart, we have some control (albeit, a soft one) over the latent space. We agree that empirically RIDE does capture quantities that we perceive as "qualitative change in the dynamics". In fact, Fig.7 and Table.2 of our work show that the $\ell_2$ distance in RIDE is large for observation pairs that we perceive as having "significantly different dynamics", such as a room with door open and closed.
>
> However, it is difficult to precisely define what having "significantly different dynamics" means, let alone giving a quantitative definition of it. The question of *why* the $\ell_2$ distance is larger for such pairs is not well-understood, and thus, requires further investigation. Without understanding *why*, we cannot guarantee that RIDE will always assign higher rewards to actions we perceive as "significantly changing the dynamics". An answer to this question of *why* will give us more confidence that the same phenomenon will be reproduced in environments with different dynamics.
>
> We also note that EC-SimCLR, our episodic memory extension of RIDE-SimCLR is able to learn good policies for the harder KeyCorridorS4R3 and KeyCorridorS5R3 environments, wheras RIDE and its episodic memory extension, RIDE-MinAgg, fails to learn any useful policies on these environments. This indicates that embeddings trained via SimCLR may be more advantageous in some environments. The results for KeyCorridorS4R3 are reported in our current version, and the results for S5R3 will be updated before the rebuttal period ends.

---

> ### Author Response · Authors · 2020-11-20
> **Author Response (part 2)**
>
> **"Are there any kinds of tasks your method might solve that would not be possible with the existing method? ... any tasks RIDE would do better at than your approach?"**
>
> We added new experiments on KeyCorridorS4R3, an environment in which RIDE is known to perform poorly [AMIGo; 2]. Although RIDE-SimCLR performs poorly in this environment as well, its episodic memory extension EC-SimCLR can learn good policies on KeyCorridorS4R3. In contrast, our episodic memory extension of RIDE, RIDE-MinAgg, fails to learn any useful policy on KeyCorridorS4R3. Additional experiments on the harder KeyCorridorS5R3 environment are currently running and will be added before the rebuttal period ends. Our preliminary results show that EC-SimCLR is able to learn good policies for this environment as well.
>
> We had initially expected RIDE to perform better on the KeyCorridor environments as they involve more interactions with objects compared to environments such as MultiRoom. However, our new experiments suggest otherwise, which contradicts our intuition. Nevertheless, we still expect RIDE to perform better on *some* environments where the agent's interaction with objects is key to solving the task.
>
> **"Less emphasis could be given to the description of RIDE vs RIDE-SimCLR's similarity to temporal distance between states."**
>
> We thank the reviewer for pointing this out. We have modified our discussion in the "Learned embedding space" section to address this comment. Specifically, we acknowledged that this is expected since RIDE clearly wasn't optimized for temporal distance, and added that this demonstrates that the intrinsic rewarding schemes of RIDE and RIDE-SimCLR, despite its apparent similarity, are qualitatively different and is not necessarily a failure of RIDE.
>
> **"For RIDE, the reward assignment feels fairly clear: taking the "open door" action creates a substantial change in RIDE's action-focused embedding space, so this is well-rewarded by RIDE."**
>
> While we do agree that this is the guiding intuition behind RIDE, we believe that a quantitative explanation for this intuition is still missing. In particular, we do observe that in the embeddings learned via forward/inverse dynamics the $\ell_2$ distance is aligned with what we perceive as "significant changes in the dynamics". However, the question of *why* $\ell_2$ distance is larger for such changes is not well-understood. As mentioned in our paper, we think quantitatively investigating this either through theory or empirical evaluation is an interesting avenue for future work.
>
> **"(RIDE-SimCLR) naturally does not provide a lot of intrinsic reward for actions but rather through movement away from where it has been, so it is natural that there is more reward for moving into new rooms rather than specifically on opening the door."**
>
> We thank the reviewer for this insightful explanation! We have updated Section 5.3 (Intrinsic Reward Visualization) to incorporate this comment.
>
> **"would have appreciated seeing a bit more analysis on running this algorithm without any external reward."**
>
> We further evaluated RIDE and RIDE-SimCLR on the first level of Mario without any external reward. The results can be found in our Appendix.
>
> **References**
>
> [1] Deepak Pathak, Pulkit Agrawal, Alexei A. Efros, Trevor Darrell. Curiosity-driven Exploration by Self-supervised Prediction. In International Conference on Machine Learning, 2017. URL: https://arxiv.org/abs/1705.05363.
>
> [2] Andres Campero, Roberta Raileanu, Heinrich Küttler, Joshua B. Tenenbaum, Tim Rocktäschel, Edward Grefenstette. Learning with AMIGo: Adversarially Motivated Intrinsic Goals. arXiv, 2020. URL: https://arxiv.org/abs/2006.12122.

---

### Official Review · AnonReviewer1 · 2020-10-27

**Rating:** 4
**Confidence:** 4

**Review:**

This paper proposes RIDE-SimCLR, that uses self-supervised learning technique like SimCLR to learn a representation for RIDE to use. The goal is to construct a representation of a state that is sensitive to their temporal distances in the trajectory. RIDE-SimCLR shows comparable performance with the original RIDE.

One main question is why adding SimCLR training is good for RL exploration? If the trajectory takes a detour to the target location, does the temporal distance make sense? Note that such a distance obviously depends on the current policy. How does the current policy affect the training? What if we use the optimal policy to train the representation? The paper doesn’t show ablation studies for that.

According to Eqn. 4, the intrinsic reward of RIDE-SimCLR is high, if the representations of two states are dissimilar with respect to each other. However, if the SimCLR training is successful, then we would expect \phi(o_i) and \phi(o_{i+1}) to be very similar (since they are 1-step away in the trajectory) and there shouldn't be any intrinsic reward? From this analysis, I feel that we might want to expect SimCLR training to be somewhat working but not fully done. But the paper doesn't really discuss these subtle cases, which is unfortunate.

Furthermore, there could be situation that two nearby state might need very different representations. E.g., before and after the agent picks the key / open the door, etc. I am a bit confused why this criterion is important. Could the author explain it?

I wonder whether the authors could give more visualization on the learned representation. Fig. 6 basically shows the SimCLR training works, which is not enough to show its usefulness. Is there any additional properties of the learned representation (e.g., sparsity? disentanglement?) How is it different from representation evaluated from RIDE other than Fig. 6?

The hyper-parameter \gamma seems to be quite important for the training, since it distinguishes between close and faraway state pairs. But we don’t see ablation studies on it. When we should set large \gamma and when to make it small?

I didn’t get what Fig. 7 shows. Where is the benefit of using RIDE-SimCLR? From the figure what part of RIDE-SimCLR that does better than RIDE?

========

I keep the score after reading the rebuttal / author comments.

I thank the authors to conduct a lot of additional experiments. Since the authors also agree that "two nearby states can have very different representations", it doesn't make sense to treat nearby states as positive pairs (and use SimCLR) to learn the representation from the beginning, which contradicts the purpose of the entire paper. If the authors cannot show other benefits of this representation learning (other than the performance boost on a 1-2 tasks), then the performance gain could be just due to other reasons that are not known.

The additional experiments (e.g., Appendix F) only compares Random with RIDE-SimCLR, and I wonder what's the performance of RIDE?

Overall the authors need to rethink the proposed approach in order to give a consistent story of what is a good representation for RL exploration.

---

> ### Author Response · Authors · 2020-11-20
> **Author Response (part 1)**
>
> We thank Reviewer1 for asking important questions and constructive criticism.
>
> **"Why (is) adding SimCLR training is good for RL exploration?"**
>
> SimCLR aids exploration by rewarding the RL agent proportional to the "effort" of reaching the encountered observation. This effort is measured in terms of the number of steps required to reach the new observation from the set of observations the agent has already encountered. Hence, it encourages the agent to explore beyond the set of observations it is familiar with (i.e., observations stored in its current episodic memory). This viewpoint has been motivated by [EC; 1] previously.
>
> **"How does the current policy affect the training? what if we use the optimal policy to train the representation?"**
>
> These are indeed interesting and important questions that would provide further insight into the learned embedding space. As the reviewer correctly noticed, the current policy affects SimCLR training by changing the notion of temporal proximity, i.e., temporal distance in SimCLR depends on the current policy. If we use the optimal policy from the beginning to train the representation, higher *intrinsic* reward will be given to observations that deviate from the optimal trajectory. However, this is not a problem if we already have a good enough policy that can reach the sparse *extrinsic* reward since the intrinsic_reward_coefficient gives more weight to extrinsic rewards than intrinsic rewards when training the policy network.
>
> The co-evolution of the policy network and the observation embedding network is extremely complex, and to the best of our knowledge, we have not seen any previous work that provides an indepth analysis of the interplay between policy networks and state representations trained online. For instance, [EC; 1] and [LaplacianRL; 2] also utilize representations trained online to compute exploration bonuses, but do not provide an indepth analysis of how the representations evolve during training. We would appreciate it if the reviewer could provide references for such works as it would provide us with additional insight.
>
> **"If the SimCLR training is successful, then we would expect \phi(o_i) and \phi(o_{i+1}) to be very similar (since they are 1-step away in the trajectory) and there shouldn't be any intrinsic reward."**
>
> We argue that this is not the case. An empirical evidene can be found in Fig.7, where the RIDE-SimCLR assigns higher rewards for passing through open doors. This is due to the specific form of the *normalized temperature-scale cross entropy* (NT-Xent) loss used in SimCLR (also defined in Section 3.1 of our paper). The reviewer is right to point out that two adjacent observations are pulled closer together by NT-Xent. However, the *normalization* in NT-Xent introduces a competition between pairs of observations within a trajectory. In particular, it pushes an observation pair farther away from each other if the temporal distance is larger than $k$ (the proximity threshold). A trajectory can be seen as a *chain* of observations which both attract and repel each other. Since the observation embeddings evolve according to both attractive and repulsive forces, the cosine distance of adjacent observation pairs do not all collapse to 0.

---

> ### Author Response · Authors · 2020-11-20
> **Author Response (part 2)**
>
> **"There could be situation that two nearby states might need very different representations."**
>
> We agree that nearby states might need very different representations. Fig.7 in our work shows that this indeed happens for RIDE-SimCLR. In particular, the agent is assigned high intrinsic reward (which corresponds to large cosine distance in the embedding space) when passing through open doors.
>
> **"More visualization on the learned representation ... any additional properties such as sparsity or disentanglement?"**
>
> In addition to Fig.6, our Fig.7 shows the intrinsic rewards given out by RIDE and RIDE-SimCLR associated with each action throughout its trajectory. We believe that these two figures sufficiently give insights on the representations (and hence the intrinsic rewards) learned via RIDE and RIDE-SimCLR, and their qualitative differenes. We would appreciate it if the reviewer could provide more details on what other visualizations would be helpful and we would gladly add them.
>
> Additional properties such as sparsity or disentaglement are not guaranteed as we did not explicitly optimize for these properties in the training. Note that properties such as sparsity or disentanglement are also not available in baselines such as RIDE, COUNT, ICM, or RND. We argue that additional properties such as sparsity or disentanglement are not necessary since we only use these learned state representations for computing intrinsic rewards, not for any other purposes. We are not sure about the benefits of having sparse representations in our setting or what kind of features should be disentangled in the observation representation space.
>
> **"hyper-parameter \gamma seems to be quite important for the training"**
>
> We do not use the hyperparameter \gamma in the training because SimCLR does not require us to explicitly sample negative pairs; explicit negative pairs are replaced by random samples from the batch in SimCLR. \gamma is used to label observation pairs in order to draw the ROC curve in Fig.6.
>
> **"I didn’t get what Fig. 7 shows. Where is the benefit of using RIDE-SimCLR?"**
>
> Fig.7 was not intended to demonstrate benefits of our approach. Rather, this visualization of intrinsic rewards is intended to highlight the qualitative difference between the RIDE and RIDE-SimCLR. It shows that, despite the apparent similarity of the two approaches, each intrinsic rewarding scheme ends up rewarding actions quite differently.
>
> **References**
>
> [1] Nikolay Savinov, Anton Raichuk, Raphaël Marinier, Damien Vincent, Marc Pollefeys, Timothy Lillicrap, Sylvain Gelly. Episodic Curiosity through Reachability. In International Conference on Learning Representations, 2019. URL: https://arxiv.org/abs/1810.02274.
>
> [2] Yifan Wu, George Tucker, Ofir Nachum. The Laplacian in RL: Learning Representations with Efficient Approximations. In International Conference on Learning Representations, 2019. URL: https://arxiv.org/abs/1810.04586.

---

### Official Review · AnonReviewer3 · 2020-10-28
**I tend to reject this paper due to (a) it relies on episodic count which does not scale to deep RL settings and (b) I expect the proposed intrinsic reward signal to be always (approximately) zero**

**Rating:** 4
**Confidence:** 4

**Review:**

The authors study exploration in procedurally-generated environments with sparse reward, proposing
a new method, termed RIDE-SimCLR, for addressing this problem. In particular, the proposed method is a direct extension of an existing method [RIDE; 1], paired with advances in contrastive unsupervised representation learning [SimCLR; 2]. Moreover, the authors form a link between RIDE-SimCLR and episodic curiosity [EC; 3] and demonstrate empirically that the proposed method is performant in procedurally-generated MiniGrid tasks.

RIDE (i.e., rewarding impact-driven exploration) is an intrinsic reward signal (Eqn. (1)) associated with the transition $o_{t} \overset{a_{t}}{\rightarrow} o_{t+1}$ given by the ratio of L2 difference in the observation embeddings, $|\| \phi(o_{t}) - \phi(o_{t+1})|\|$,  over the (square root of the) episodic count for $o_{t+1}$. The main contribution of this paper is to reconsider what the embedding function $\phi$ should be such that the numerator (or the distance between two observation embeddings) has an explicit conceptual meaning, i.e., represent a clear "measure of impact. The authors borrow the idea from SimCLR [2] and use cosine similarity between successive observation embeddings as a measure of impact when the embedding function $\phi$ is trained with contrastive learning, treating observations that $k$-steps reachable as positive pairs, i.e., forcing their embeddings to be approximately equal.

Moreover, the authors the draw connections between their proposed method, RIDE-SimCLR, and EC [3]. I find this formulation interesting and useful.

**Questions and Concerns**:
1. Both RIDE and RIDE-SimCLR use episodic count. This is trivial to calculate in MiniGrid (as it's given by default) but it is not accessible in more complicated settings (e.g., what about a procedurally generated robot arm manipulation task?). Also it's unclear to me if authors use the episodic of observations or simulator states! The former is prone to fail when there are aliased observations, i.e., the same observation in different states. If authors use simulator state visitation/episodic count that means that they access privileged information and hence all the RIDE results from the original paper should be revisited since the comparison to the baselines (ICM, RND, Count) is unfair. Most notably, if in the NoisyTV task they use simulator state count then this is indifferent for RIDE and hence RIDE-SimCLR. Pseudo-counts [4] can be used instead but this comes with all the issues count-based exploration has in deep RL [4], e.g., noisy TV etc.
2. By construction of $\phi$ in RIDE-SimCLR, shouldn't (asymptotically at least) $1 - \text{cos}(\phi(o_{t}), \phi(o_{t+1})) = 0$ for all $t$ since $\Delta t(o_{t+1}, o_{t}) = 1 < k = 2$? That should mean that if $\phi$ is doing what it's trained to do, then $R_{\text{SimCLR}} = 0$ always and then there is no intrinsic reward. I am not sure if I am interpreting something incorrectly or if there is some artifact due to the denominator term (i.e., episodic count) which confounds with the empirical results we see, since I would expect the numerator in Eqn. (4) to be approximately zero.

**Notes**:
1. Make sure that the RIDE definition in Eqn. (1) makes sense, i.e., $s_{t}$ and $a_{t}$ are in the middle equality and $o_{t}$, $o_{t+1}$ in the RHS. Same applies to Eqn. (4).
2. It would be really useful to see the methods in Section 3.2 summarised in a table with columns: (i) method; (ii) $\phi$ trained via; (iii) aggregator $A$.

**References**

[1] Roberta Raileanu and Tim Rockt¨aschel. Ride: Rewarding impact-driven exploration for
procedurally-generated environments. In International Conference on Learning Representations,
2020. URL https://openreview.net/forum?id=rkg-TJBFPB.

[2] Ting Chen, Simon Kornblith, Mohammad Norouzi, and Geoffrey Hinton. A simple framework for
contrastive learning of visual representations. arXiv preprint arXiv:2002.05709, 2020.

[3] Nikolay Savinov, Anton Raichuk, Rapha¨el Marinier, Damien Vincent, Marc Pollefeys, Timothy
Lillicrap, and Sylvain Gelly. Episodic curiosity through reachability. In International Conference
on Learning Representations (ICLR), 2019.

[4] Marc Bellemare, Sriram Srinivasan, Georg Ostrovski, Tom Schaul, David Saxton, and Remi Munos.
Unifying count-based exploration and intrinsic motivation. In Advances in Neural Information
Processing Systems, pp. 1471–1479, 2016.

---

> ### Author Response · Authors · 2020-11-20
> **Author Response**
>
> We thank Reviewer3 for pointing out an important mistake in our write-up and asking thought-provoking questions. To address the reviewer's comments, we updated the submission to 1) correct equations regarding $s_t$ and $o_t$, 2) summarize the methods in a table. We acknowledge that 1) is a non-negligible change and that it may affect the reviewer's assessment of our method since the simulator state count used in RIDE (and RIDE-SimCLR) is privileged information that cannot be accessed by ICM or RND. We expand on this issue of fairness when addressing the reviewer's **"Comparison to the baselines (ICM, RND, Count) is unfair"** comment.
>
> **"Observation count or simulator state count?"**
>
> We used simulator states (more precisely, the full state of the MiniGrid) for the episodic state visitation counts. We noticed that we incorrectly wrote $N(o_{t+1})$ instead of $N(s_{t+1})$ in eqn (1). We updated our submission to correct this mistake.
>
> **"Comparison to the baselines (ICM, RND, Count) is unfair."**
>
> As the reviewer correctly pointed out, episodic simulator state count is indeed privileged information not made available to ICM or RND (Count still uses the simulator state count in [RIDE;1]). Hence, there may be an issue of fairness when comparing RIDE to ICM or RND on MiniGrid. However, the experiments and findings of our work remain valid since we are *not* trying to claim the state-of-the-art amongst *all* exploration methods. Our focus is on the comparison to RIDE. In particular, we address the question of whether the forward/inverse dynamics training can be replaced with a different, more interpretable training objective while retaining RIDE's strong performance on the procedurally-generated MiniGrid environments. As a side note, EC-SimCLR does not use episodic count since any repeated *observation* will give 0 intrinsic reward.
>
> **"If in the NoisyTV task they use simulator state count then this is indifferent for RIDE and hence RIDE-SimCLR."**
>
> We do not believe that episodic count will make RIDE (and RIDE-SimCLR) indifferrent to the Noisy TV. A "switch channel" action by the agent will change the color of the TV, which will in turn change the state (because of the TV's color). Hence, the changed state will be tallied as a distinct state in the episodic count.
>
> **"Episodic count does not scale to deep RL settings."**
>
> There are several ways to scale-up RIDE-SimCLR and EC-SimCLR to larger (possibly continuous) state spaces. For EC-SimCLR, one can start with a fixed-size episodic memory buffer and add a new observation to memory only if the cosine distance from observations contained in the buffer is over a certain threshold, as proposed in [EC; 2]. If the buffer reaches capacity, then one can randomly replace an observation in the memory with the newly committed observation. For RIDE-SimCLR, one can use the approximate state visitation count proposed in [NGU; 3] (defined in eqn. 2 of the paper), which uses the sum of similarities (computed with a kernel function) to the observations in episodic memory as a proxy to state counts for continuous states.
>
> **"Asymptotically, cos goes to 0?"**
>
> No, the cosine distance does not go to 0. This is due to the specific form of the *normalized temperature-scale cross entropy* (NT-Xent) loss used in SimCLR (also defined in Section 3.1 of our paper). The reviewer is right to point out that two adjacent observations are pulled closer together by NT-Xent. However, the *normalization* in NT-Xent introduces a competition between pairs of observations within a trajectory. In particular, it pushes an observation pair farther away from each other if the temporal distance is larger than $k$ (the proximity threshold). A trajectory can be seen as a *chain* of observations which both attract and repel each other. Since the observation embeddings evolve according to both attractive and repulsive forces, the cosine distance of adjacent observation pairs do not all collapse to 0. We can also provide a plot of the RIDE-SimCLR intrinsic reward throughout the training, if the reviewer requests it.
>
> **References**
>
> [1] Roberta Raileanu and Tim Rocktäschel. Ride: Rewarding impact-driven exploration for procedurally-generated environments. In International Conference on Learning Representations, 2020. URL: https://arxiv.org/abs/2002.12292.
>
> [2] Nikolay Savinov, Anton Raichuk, Raphaël Marinier, Damien Vincent, Marc Pollefeys, Timothy Lillicrap, Sylvain Gelly. Episodic Curiosity through Reachability. In International Conference on Learning Representations, 2019. URL: https://arxiv.org/abs/1810.02274.
>
> [3] Adrià Puigdomènech Badia, Pablo Sprechmann, Alex Vitvitskyi, Daniel Guo, Bilal Piot, Steven Kapturowski, Olivier Tieleman, Martín Arjovsky, Alexander Pritzel, Andew Bolt, Charles Blundell. Never Give Up: Learning Directed Exploration Strategies. In International Conference on Learning Representations, 2020. URL: https://arxiv.org/abs/2002.06038.

---

> > ### Comment · AnonReviewer3 · 2020-11-23
> > **Response to Authors' Rebuttal**
> >
> > Thank you for your reply. Happy I could help with the state/observation count confusion.  A few more comments on your response:
> >
> > > However, the experiments and findings of our work remain valid since we are not trying to claim the state-of-the-art amongst all exploration methods. Our focus is on the comparison to RIDE. In particular, we address the question of whether the forward/inverse dynamics training can be replaced with a different, more interpretable training objective while retaining RIDE's strong performance on the procedurally-generated MiniGrid environments.
> >
> > I cannot fully agree with this statement. If you accept that the method you build on (i.e., RIDE) is based on weak foundations (i.e., it relies on privileged information that other methods do not require) then I do not see why making a more interpretable version of it while accepting its limited applicability adds value.
> >
> > > There are several ways to scale-up RIDE-SimCLR and EC-SimCLR to larger (possibly continuous) state spaces. For EC-SimCLR, one can start with a fixed-size episodic memory buffer and add a new observation to memory only if the cosine distance from observations contained in the buffer is over a certain threshold, as proposed in [EC; 2]. If the buffer reaches capacity, then one can randomly replace an observation in the memory with the newly committed observation. For RIDE-SimCLR, one can use the approximate state visitation count proposed in [NGU; 3] (defined in eqn. 2 of the paper), which uses the sum of similarities (computed with a kernel function) to the observations in episodic memory as a proxy to state counts for continuous states.
> >
> > I do agree with your suggestions in scaling your method in deep RL, however, I struggle to see what's the novelty of it in this case, since your suggestion is really to combine "tricks" from EC [2] and NGU [3]. I do appreciate that EC-SimCLR does not require a comparator network, while both EC and NGU do but to convince myself that you can do without a comparator network by using SimCLR's cosine similarity and the reachability graph, I would at least need to see some qualitative and quantitative comparison with EC and NGU. Having said that:
> >
> > > A trajectory can be seen as a chain of observations which both attract and repel each other. Since the observation embeddings evolve according to both attractive and repulsive forces, the cosine distance of adjacent observation pairs do not all collapse to 0.
> >
> > I do see why in practice the cosine similarity won't be equal **exactly** to 1 and hence the intrinsic reward won't be exactly equal to 0 and how the temperature (i.e., calibration) parameter can amplify this gap but my concern is that by definition there is a contradiction, where RIDE-SimCLR selects successive states that have a higher intrinsic reward while the embedding loss pushes all successive states to have almost 1 cosine similarity. I do believe that the overall pipeline **may** work because your embedding network fails to pull together pairs of observations that you have not seen (i.e., fails to generalise to new observations) and you end up seeking these states, which of course is a good idea, similar to what the RND is doing but it is rather different from what you suggest the method is doing. I guess an easy way of showing that is by (a) fixing a policy; (b) fixing the environment (the layout too); (c) training your embedding network, by exhaustively sampling the states in this environment and (d) running RIDE-SimCLR in the environment and seeing what the **numerator** of RIDE-SimCLR is in all the states. I expect this to be a constant term, which depends on the overall error of your embedding network -- if you provide it with enough capacity I expect the numerator of the intrinsic reward to be **almost** 0 everywhere.

---

> > > ### Author Response · Authors · 2020-11-24
> > > **Author Response**
> > >
> > > We thank Reviewer3 for more insightful comments.
> > >
> > > **"If you accept that the method you build on (i.e., RIDE) is based on weak foundations (i.e., it relies on privileged information that other methods do not require) then I do not see why making a more interpretable version of it while accepting its limited applicability adds value."**
> > >
> > > We would like to point out that our episodic memory extensions, RIDE-MinAgg and EC-SimCLR, do not require this privileged state count, and outperform RIDE and RIDE-SimCLR.
> > >
> > > Moreover, we ran additional experiments on KeyCorridorS4R3 and KeyCorridorS5R3 (now updated in Fig.5), and show that EC-SimCLR, our episodic memory extension of RIDE-SimCLR is the only method that can learn good policies for these harder environments. In contrast, RIDE and its episodic memory extension, RIDE-MinAgg, fail to learn any useful policies on these environments. This indicates that embeddings trained via SimCLR may be more advantageous in some environments, in addition to being more interpretable.
> > >
> > > **"I do agree with your suggestions in scaling your method in deep RL ... (but) to convince myself that you can do without a comparator network by using SimCLR's cosine similarity and the reachability graph, I would at least need to see some qualitative and quantitative comparison with EC and NGU."**
> > >
> > > We agree with the reviewer that to demonstrate the utility of SimCLR in scaled-up RL environments, more evaluations and baselines are needed. However, we would like to note that the primary focus of our work was to address the question of whether the forward/inverse dynamics training of RIDE can be replaced with a more interpretable training objective while retaining RIDE's strong performance on the current benchmarks.
> > >
> > > **"My concern is that by definition there is a contradiction, where RIDE-SimCLR selects successive states that have a higher intrinsic reward while the embedding loss pushes all successive states to have almost 1 cosine similarity."**
> > >
> > > We apologize for misreading the reviewer's comment. We initially read the reviewer's comments too literally and failed to address the real underlying question. The reviewer's comment, *"the overall pipeline **may** work because your embedding network fails to pull together pairs of observations that you have not seen (i.e., fails to generalise to new observations) and you end up seeking these states, which of course is a good idea, similar to what the RND is doing."* is spot-on. We agree that if we train the embedding network using a fixed, deterministic policy on the same environment configuration, the cosine similarities of successive observations may converge to some constant.
> > >
> > > However, the observation embeddings are trained *online* with the policy network. Hence, the policy network dynamically evolves along with the observation embedding network. The changing policy supplies the embedding network with new pairs of observation, and the intrinsic reward from the embeddings in turn pushes the policy towards more uncharted territories. Once the agent's exploration stumbles upon the extrinsic reward, weight coefficients of the extrinsic and intrinsic reward will control the exploration-exploitation trade-off. If the agent succeeds in learning an optimal policy that will guide it to the sparse extrinsic reward, then it matters less whether the intrinsic reward encourages deviation from the optimal trajectory since the extrinsic reward is given much higher weight. We added a plot (Fig.10) in Appendix G to demonstrate that average intrinsic reward (which corresponds to cosine distance) during training does not go to 0.

---

### Official Review · AnonReviewer2 · 2020-10-28
**Interesting idea and well-explained chain of logic, but could use better benchmarking and**

**Rating:** 4
**Confidence:** 3

**Review:**

This work explores novel intrinsic motivations used to augment extrinsic rewards in sparse reward contexts. Their algorithmic contributions are essentially two-fold:
-They propose RIDE-SimCLR, a modification of RIDE which replaces RIDE's intrinsic motivation (the l2-distance, between timesteps, of an embedding of observations) with a SimCLR-inspired contrastive learning dissimilarity measure. This is claimed to be a more conceptually clear/natural choice.
-They integrate a version of episodic memory with this, proposing EC-SimCLR, with an episodic RIDE ablation RIDE-MinAgg.
Using procedurally-generated enviuronments from MiniGrid, they go on to show superior performance of EC-SimCLR over RIDE, RIDE-SimCLR, and RIDE-MinAgg, while they claim RIDE and RIDE-SimCLR to have comparable performance.

This paper does a clear job of motivating the steps of the innovations described above, and its experimental demonstrations show superior performance of EC-SimCLR over reasonable ablations of these innovations. Its technical background pieces are lucid, walking the reader through several important concepts.

I see two key weaknesses:

1) The set of environments and baselines is fairly narrow. They choose to focus on procedurally-generated MiniGrid environments, for which the RIDE paper demonstrates superiority over a number of other intrinsic motivation methods, and hence they exclude these other methods in evaluations. RIDE's evaulations on MiniGrid appears more extensive (this is only a subset of these environments, correct?), and its evaluations on other environments show less of a clear advantage over these other methods. While generally we cannot expect researchers to run their algorithms on all of our favorite environments -- that is a lot of work! -- it would be much more convincing if the authors compared their algorithm to a broader list of baselines on an environment for which RIDE did not show such favorable performance, given the similarity of their approach to that of RIDE's. I do not agree with the statement: "We note that MiniGrid provides a sufficiently challenging suite of tasks for RL agents despite its apparent simplicity, as ICM and RND fail to learn any effective policies for
some tasks due to the difficulty posed by procedurally-generated environments." When useful baselines fail on a certain subset of tasks, it does not mean that this subset of tasks is the suite of tasks with which to judge future algorithms.

2) A good deal of the technical treatment is devoted to motivating's the innovation of RIDE-SimCLR over RIDE as being more conceptually clear/natural. This argument seems opaque to me. As evidence, they show that RIDE-SimCLR's similarity metric much better predicts temporal difference in states than RIDE's l2 distance. This is, however, entirely expected given the objective of the similarity metric, and is it not case that RIDE's l2 distance might have some conceptually clear/natural interpretation?  Hence this argument of conceptual clarity seems a bit soft, despite quite a bit of the exposition devoted to it. If there were clearer utilities demonstrated, it would be more convincing, but right now I find this emphasis confusing.

This innovation does appear to have one utility: that it suggests the further improvement that involves episodic memory (though, is this the only route towards this further improvement?).

I recommend rejection, though I am certainly open to changing my mind, especially if the case can be made that the benchmarks are exhaustive enough, or that it's unreasonable to expect more benchmarking within a single publication.

A question for authors: unless I misunderstood, EC itself was not used as a baseline. Why is that?

One minor point: I think the definition of EC-SimCLR could be spelled out more explicitly. I think I know what it is, but it is not "RIDE-SimCLR with A = min" given that RIDE-SimCLR has its own specific A.

---

> ### Author Response · Authors · 2020-11-20
> **Author Response (part 1)**
>
> We thank Reviewer2 for helpful comments and suggestions. We are glad that the reviewer found our motivation and exposition lucid.
>
> **"Set of envs and baselines is fairly narrow"**
>
> We certainly agree that RIDE had more extensive evaluations. We decided to subsample a representative set of environments from ones considered in [RIDE; 1] for two reasons: 1) we wanted our work to serve as a *proof-of-concept* that forward/inverse dynamics training in RIDE can be replaced with a different training objective that retains RIDE's strong performance on MiniGrid and yields an interpretable intrinsic reward, and 2) we had limited compute.
>
> Of course, if one were to claim the state-of-the-art amongst *all* exploration methods, extensive evaluations over multiple environments, baselines, and hyperparameter configurations are necessary irrespective of one's compute resources. This brings us to our 1st reason: we are not trying to claim the state-of-the-art. Our work focuses on the comparison to RIDE and addresses the question of whether the forward/inverse dynamics training can be replaced with a different, more interpretable training objective while retaining RIDE's strong performance on the procedurally-generated MiniGrid environments. Hence, we use only RIDE as the baseline. Please also note that the performance of other baselines on the environments we evaluate is reported in [RIDE; 1] and [AMIGo; 2]. Hence, our comparison to RIDE essentially covers other baselines as well.
>
> Regarding our 2nd reason (limited compute), [RIDE; 1] ran a grid search over 3 x 2 x 4 x 7 x 6 = 1008 (learning_rate, batch_size, unroll_length, intrinsic_reward_coefficient, entropy_coefficient) hyperparameter configurations to obtain the best possible hyperparameters for *each* model. Given our CPU/GPU settings, it took us at least 6hrs to train until convergence on a single hyperparameter configuration. Hence, about 6,000 hrs of dedicated compute would be required for *each* model on *each* new environment to perform exhaustive search at the level of [RIDE; 1]. Instead, we chose to capitalize on the extensive search performed in [RIDE; 1] by restricting our evaluation to MiniGrid environments.
>
> We chose MultiRoomN7S8 over N7S4, and N12S10 over N10S4 and N10S10 because we believe that larger number of rooms and larger room sizes make the MultiRoom environment more difficult. This belief is in accordance with Fig.3 in [RIDE;1] and Fig.5 in our work, where we roughly measure difficulty by the number of frames required to learn a good policy.
>
> To address the reviewer's comment that the set of environments is fairly narrow, we ran additional experiments on KeyCorridorS4R3, KeyCorridorS5R3, and Mario. We expand on this in the next paragraph.
>
> **"It would be much more convincing if broader list of baselines on an environment for which RIDE did not show favorable performance."**
>
> To address this comment, we ran additional experiments on 1) KeyCorridorS4R3 and 2) first level of SuperMarioBros without any extrinsic reward and reported the results in our Experiments section (KeyCorridorS4R3) and the Appendix (Mario). We are currently running experiments on KeyCorridorS5R3 and will report the results before the rebuttal period ends.
>
> 1) We chose KeyCorridorS4R3 since it has been experimentally shown in [AMIGo; 2] that RIDE fails to learn a good policy for this environment even after training on 5e8 frames. Our experimental results show that EC-SimCLR is able to learn a good policy for KeyCorridorS4R3 sample-efficiently (in fact, our preliminary results show that EC-SimCLR is able to learn a good policy even for the harder KeyCorridorS5R3 environment). However, RIDE-SimCLR and RIDE-MinAgg fail to learn a good policy for this task. We thank the reviewer for encouraging us to explore harder environments that RIDE didn't show much favorable performance in.
>
> 2) We chose SuperMarioBros (without extrinsic reward) to show that RIDE-SimCLR can still explore effectively without extrinsic reward in non-MiniGrid environment. For the SuperMarioBros environment, RIDE did not perform much better compared to other intrinsic rewards. In fact, it is noted in [RIDE; 1] that even IMPALA, which doesn't use intrinsic rewards, performs similar to other exploration methods such as RIDE, ICM, RND, and COUNT, hinting that this environment was not that hard to begin with. Note that this environment is not procedurally-generated; it is a single-seed environment.

---

> ### Author Response · Authors · 2020-11-20
> **Author Response (part 2)**
>
> **"Motivating the innovation of RIDE-SimCLR over RIDE as being more conceptually clear/natural seems opaque to me ... if there were clearer utilities demonstrated, it would be more convincing."**
>
> We have updated our exposition in Section 2.1. "Impact-driven Exploration" to add clarity to our central argument. In particular, we point out that RIDE's training objective does not explicitly pull together or push apart embeddings of *different* observation pairs ($o_i, o_j$). To elaborate, in [ICM; 3], the $\ell_2$ distance between $\phi(o_{t+1})$ and the prediction $\hat{\phi}(o_{t+1})$ has a clear interpretation as the forward prediction "error", since the forward dynamics objective explicitly minimizes this $\ell_2$ distance. RIDE, on the other hand, uses the $\ell_2$ distance between *different* observations $o_t$ and $o_{t+1}$ as the intrinsic reward. Yet, the forward and inverse dynamics objective does not specify which pairs of observation embeddings ($o_i$,$o_j$) should be pulled closer together and which should be pushed apart (in $\ell_2$ distance).
>
> However, this does not mean that RIDE fails to capture qualitative change in the dynamics. In fact, Fig.7 and Table.2 of our work show that the $\ell_2$ distance in RIDE is large for observation pairs that we perceive as having "significantly different dynamics", such as a room with door open and closed. However, it is difficult to precisely define what having "different dynamics" means, let alone giving a quantitative definition of it. Moreover, the question of *why* the $\ell_2$ distance is larger for such pairs is not well-understood, and thus, requires further investigation. Without understanding *why*, we cannot guarantee that RIDE will always assign higher rewards to actions we perceive as "significantly changing the dynamics". An answer to this question of *why* will give us more confidence that the same phenomenon will be reproduced in environments with different dynamics. Therefore, we argue that interpretability for intrinsic rewards is itself a clear utility.
>
> We also note that EC-SimCLR, our episodic memory extension of RIDE-SimCLR is able to learn good policies for the harder KeyCorridorS4R3 and KeyCorridorS5R3 environments, wheras RIDE and its episodic memory extension, RIDE-MinAgg, fails to learn any useful policies on these environments. This indicates that embeddings trained via SimCLR may be more advantageous in some environments.
>
> **"EC itself was not used as a baseline. Why is that?"**
>
> We did not consider EC as a baseline because wanted to focus the narrative on the comparison to RIDE. Our main goal was to retain the strong performance of RIDE on procedurally-generated environments like MiniGrid while getting a well-defined similarity measure for state representations (and hence, more interpretable intrinsic reward). Comparing EC-SimCLR and EC [EC; 4] to tease apart the effect of the relatively recent SimCLR training is indeed an interersting question, but we consider this to be outside the scope of this work.
>
> **"Definition of EC-SimCLR could be spelled out more explicitly."**
>
> We updated our Section 3.2 to define EC-SimCLR more explicitly. In addition, we added Table 1 to give a summary of all the models considered in our work (RIDE, RIDE-MinAgg, RIDE-SimCLR, EC-SimCLR).
>
> **References**
>
> [1] Roberta Raileanu and Tim Rocktäschel. Ride: Rewarding impact-driven exploration for procedurally-generated environments. In International Conference on Learning Representations, 2020. URL: https://arxiv.org/abs/2002.12292.
>
> [2] Andres Campero, Roberta Raileanu, Heinrich Küttler, Joshua B. Tenenbaum, Tim Rocktäschel, Edward Grefenstette. Learning with AMIGo: Adversarially Motivated Intrinsic Goals. arXiv, 2020. URL: https://arxiv.org/abs/2006.12122.
>
> [3] Deepak Pathak, Pulkit Agrawal, Alexei A. Efros, Trevor Darrell. Curiosity-driven Exploration by Self-supervised Prediction. In International Conference on Machine Learning, 2017. URL: https://arxiv.org/abs/1705.05363.
>
> [4] Nikolay Savinov, Anton Raichuk, Raphaël Marinier, Damien Vincent, Marc Pollefeys, Timothy Lillicrap, Sylvain Gelly. Episodic Curiosity through Reachability. In International Conference on Learning Representations, 2019. URL: https://arxiv.org/abs/1810.02274.

---

> > ### Comment · AnonReviewer2 · 2020-11-23
> > **Thanks for the detailed responses! A few concerns remain.**
> >
> > Thank you for the reply. The additional experiments run seem useful -- in particular the KeyCorridorS4R3 result is nice. I agree that the SuperMarioBros result might not be super interpretable, as it simply may not demand much of models. And I'm certainly sympathetic to constraints on compute! One shouldn't be expected to run every reviewer's favorite environment.
> >
> > A couple remaining concerns.
> >
> > Re: Section 2.1 additions, if I am understanding the current argument correctly, you perceive RIDE to have a potential deficit (the right sorts of successive states being far apart in embedding space) because a lack of deficit isn't apparent from the definition. But do you know that failure cases arise as a result? We might not understand the "why" of whether something works or not, and hence fear that it might not, but without empirically demonstrating the failure, it's hard to say that this has that sort of deficit!
> >
> > Re: EC not being included as a baseline, to me this still seems like an odd choice. EC-SimCLR seems to be your best-performing model (by quite a bit, after this recent addition) and so the ablation analysis that makes the most sense to me is that which figures out what components of it are essential. Why focus on a comparison to RIDE?

---

> > > ### Author Response · Authors · 2020-11-24
> > > **Author Response**
> > >
> > > We greatly appreciate the reviewer's understanding and helpful comments.
> > >
> > > **"We might not understand the "why" of whether something works or not, and hence fear that it might not, but without empirically demonstrating the failure, it's hard to say that this has that sort of deficit!"**
> > >
> > > This is a valid concern and we are glad the reviewer raised this point. To zoom in on the failure mode of RIDE, we visualized the intrinsic rewards of a RIDE agent trained on the KeyCorridorS4R3 environment (Fig.11 of Appendix H). Note that RIDE *failed* to learn a good policy for this hard environment. We observed that the "open door" action wasn't rewarded as highly as in the MultiRoomN7S8 environment, in which the RIDE agent learned a good policy. Instead, the seemingly insignificant "turn left/right" actions were highly rewarded, which demonstrates a failure mode of RIDE's embedding training.
> > >
> > > **"EC not being included as a baseline, to me this still seems like an odd choice ... the ablation analysis that makes the most sense to me is that which figures out what components of it are essential. Why focus on a comparison to RIDE?"**
> > >
> > > In light of the new results on KeyCorridorS4R3 and KeyCorridorS5R3, this is a valid point. A comparison to EC indeed seems necessary to determine what components of EC-SimCLR are necessary for such performance. However, our initial focus was not on proposing a new state-of-the-art exploration method. Instead, we had a more modest goal of replacing the embedding training component of RIDE with a more explicit and interpretable procedure while retaining RIDE's performance. We were mainly concerned with the conceptual difficulties RIDE's intrinsic reward posed (using $\ell_2$ distance in a space that was not explicitly trained with this distance measure) and wanted to address this. Moreover, we proposed episodic memory extensions, EC-SimCLR and RIDE-MinAgg, simply to demonstrate the usefulness of the connection we make between RIDE and EC.
> > >
> > > We admit that EC-SimCLR exceeded our initial expectations by being the only method (among the ones we evaluate) that successfully learns both KeyCorridorS4R3 and KeyCorridorS5R3, whereas RIDE and its episodic memory extension fail to learn any useful policy for the easier KeyCorridorS4R3 environment. And we agree that the natural next step is a comparison to EC on these environments to concretely determine where EC-SimCLR stands in comparison to other state-of-the-art exploration methods.

---

### Author Response · Authors · 2020-11-25
**Summary of Updates**

We thank all the reviewers for helpful comments and questions. Based on the feedback we received, we made the following updates to our submission.

- Based on comments from Reviewer4, we updated our exposition on RIDE in Section 2.1. We also added a new paragraph in Section 2.1 to clarify our central argument regarding RIDE and interpretability in response to overall feedback from the reviewers.
- Based on comments from Reviewer3, we corrected eq.1 in Section 2.1 to show that state visitation counts are used to discount the RIDE/RIDE-SimCLR rewards.
- Based on comments from Reviewer2, we added a table summarizing all the methods we evaluate in this paper at the end of Section 3.2 (Table 1).
- In response to comments from Reviewer2 and Reviewer4, we ran additional experiments on harder environments (KeyCorridorS4R3/S5R3) and updated Section 5 (Fig.5). The results show that EC-SimCLR is the only method (among the ones we evaluate) that successfully learns both KeyCorridorS4R3 and KeyCorridorS5R3, indicating that the embeddings trained via SimCLR may be advantageous in some environments in addition to being more interpretable.
- In response to comments from Reviewer2, we ran experiments on Mario without extrinsic reward to show that RIDE-SimCLR can still explore effectively without extrinsic reward in non-MiniGrid environments. The results can be found in Fig.9 of Appendix F.
- In response to comments from Reviewer1 and Reviewer3, we added a plot (Fig.10) in Appendix G to demonstrate that average intrinsic reward (which corresponds to average cosine distance) during training does not go to 0.
- In response to comments from Reviewer2, we visualized the intrinsic rewards of a RIDE agent trained on the KeyCorridorS4R3 environment in Fig.11 of Appendix H to zoom in on a failure mode of RIDE's embedding training.

---

### Decision · Program_Chairs · 2021-01-07
**Final Decision**

**Decision:**

Reject

**Comment:**

I thank the authors for their submission and very active participation in the author response period. I want to start by stating that I rank the paper higher as is currently reflected in the average score of the reviewers. The reasons for this are that a) R2 and R3, while responding to the author's rebuttal, do not seem to have updated their score or indicated that they want to keep their initial assessment of the paper -- in particular, R2 has acknowledged that additional experiments by the authors were useful and results on KeyCorridorS4/S5R3 are nice, and b) I disagree with R2's sentiment that MiniGrid is not a suitable testbed -- it is by now an established benchmark for evaluating RL exploration and representation learning methods (see list of publications on https://github.com/maximecb/gym-minigrid). However, despite my more positive stance on the paper, I fully agree with R1 and R2 that a comparison to EC is needed in order to shed light into which factors of EC-SimCLR actually led to improvements in comparison to RIDE. I therefore recommend rejection, but I strongly encourage the authors to take the feedback from the reviewers and work on a revised submission to the next venue.